# Expression and Kinetics of Endogenous Cannabinoids in the Brain and Spinal Cord of a Spare Nerve Injury (SNI) Model of Neuropathic Pain

**DOI:** 10.3390/cells11244130

**Published:** 2022-12-19

**Authors:** Kenta Kurosu, Ariful Islam, Tomohito Sato, Tomoaki Kahyo, Tomohiro Banno, Noriko Sato, Yukihiro Matsuyama, Mitsutoshi Setou

**Affiliations:** 1Department of Orthopedic Surgery, Hamamatsu University School of Medicine, Hamamatsu 431-3192, Shizuoka, Japan; 2Department of Cellular and Molecular Anatomy, Hamamatsu University School of Medicine, Hamamatsu 431-3192, Shizuoka, Japan; 3International Mass Imaging Center, Hamamatsu University School of Medicine, Hamamatsu 431-3192, Shizuoka, Japan; 4Department of Systems Molecular Anatomy, Institute for Medical Photonics Research, Preeminent Medical Photonics Education & Research Center, Hamamatsu 431-3192, Shizuoka, Japan

**Keywords:** endogenous cannabinoids, 2-arachidonoyl glycerol, DESI-MSI, spare nerve injury, periaqueductal gray, hypothalamus, lumbar spinal cord

## Abstract

The role of endogenous cannabinoids in neuropathic pain has been actively studied, among which 2-arachidonoyl glycerol (2-AG) has received the most attention. However, owing to its chemical properties, direct detection of 2-AG distribution in tissues is difficult. Moreover, although desorption electrospray ionization mass spectrometry imaging (DESI-MSI) has enabled the detection of 2-AG, its distribution in the brain and spinal cord of neuropathic pain models has not been reported. In this study, the expression and distribution of 2-AG in the brain and spinal cord of a spare nerve injury (SNI) mice model of neuropathic pain was examined using DESI-MSI. The brain and lumbar spinal cord were collected and analyzed on days 3, 7, and 21 after treatment. On days 3 and 7 after treatment, 2-AG expression in the SNI model was decreased in the hypothalamus, midbrain, and especially in the periaqueductal gray (PAG) region but increased in the lumbar spinal cord. On day 21, the SNI model showed decreased 2-AG expression in the hypothalamus, but the difference from the control was not significant. Furthermore, there were no differences in 2-AG expression between the lumbar spinal cord, midbrain, or PAG. These data suggest that 2-AG might be involved in pain control.

## 1. Introduction

Research confirmed that cannabis-induced psychiatric symptoms were primarily caused by the fat-soluble Δ9-tetrahydrocannabinol (THC) and cannabinoid receptors which were identified in 1990 [1]. Two types of cannabinoid receptors have been identified so far (CB1 and CB2), but around that time, the search had begun for biological marijuana analogs (endogenous cannabinoids), and as a result, *N*-arachidonoylethanolamine (anandamide: AEA) was found in 1992 [2]. Subsequently, in 1995, 2-arachidonoylglycerol (2-AG) was identified as an endogenous cannabinoid [3,4]. All these endogenous cannabinoids are neutral lipids containing arachidonic acid. Initially, anandamide was studied first, but 2-AG was found to be 50–200 times more abundantly expressed in the brain than anandamide [5]. Because it is a full agonist of the CB1 receptor, 2-AG is considered as a true bioactive substance. The 2-AG is also synthesized from phospholipid precursors in postsynaptic neurons by the activation of phospholipase C (PLC) and diacylglycerol lipase (DAGL). Subsequently, when 2-AG is released at the synaptic cleft, it returns to the cell and is degraded by hydrolytic enzymes such as fatty acid amide hydrolase (FAAH) and monoacylglycerol lipase (MAGL) present in the presynaptic cell. Thus, research on 2-AG is progressing. In addition, it was reported that 2-AG could be detected with high sensitivity by desorption electrospray ionization mass spectrometry imaging (DESI-MSI) [6], and the intra-tissue distribution of 2-AG has been investigated in detail. Islam A et al. reported that water-immersion stress increased 2-AG distribution in the hypothalamus, hindbrain, and midbrain of SAMP 8 mice brain [7]. Although the distribution of 2-AG in the brain due to stress response has been described, the expression and distribution of 2-AG in the brain or other tissues due to other stresses are still unknown.

There have been numerous reports of the distribution of endogenous cannabinoids in various mouse models of pain. Elevated spinal cord levels of endogenous cannabinoids have been reported in rat models of acute and chronic pain [8,9,10]. Wotherspoon G et al. reported that in a spinal nerve ligation model rat with partial mechanical injury of the sciatic nerve, the level of CB2 receptor protein was upregulated in the injured spinal cord. They also reported increased expression of CB2 receptor protein in the spinal cord and dorsal root ganglion in a rat model of sciatic nerve transection [11]. Furthermore, in a chronic constriction injury (CCI) rat model of sciatic nerve stenosis, elevated expression of CB1 receptor protein and CB2 receptor-related mRNA was reported in injured spinal cords [12,13]. In addition, CB1 receptor protein was upregulated in the injured spinal cord of a spare nerve injury (SNI) rat model [14], in which partial sciatic nerve ligation causes degeneration of the dorsal horn of the spinal cord, resulting in abnormal pain [15]. However, these studies did not examine the actual dynamics or distribution of 2-AG, which is thought to have physiological activity. Thus, the function of 2-AG in SNI remains unclear.

Therefore, the purpose of this study was to investigate in detail the dynamics and distribution of 2-AG in the brain and spinal cord of an SNI mouse model of neuropathic pain using DESI-MSI.

## 2. Materials and Methods

### 2.1. Animals

Eight-week-old male C57BL/6JJmsSlc mice (16–21 g) purchased from SLC Inc. (Hamamatsu, Japan) were used in this study. The mice were monitored at least once per day. They were housed under controlled conditions of temperature (23 ± 1 °C), relative humidity (50% ± 10%), and 12-h light/dark cycle (lights on from 7 AM to 7 PM), with free access to food and water. We established early/humane endpoints as follows: weight loss greater than 20% in a day as an indicator of distress or suffering in rodents, arched position and prominence of the spine as a sign of poor body condition/sickness, excessive grooming or flinching, vocalization, and torpor. If one animal displayed one or more of the endpoints during the daily checking, we would euthanize it immediately.

SNI models were performed as described in Decosterd I and Woolf CJ [14]. Briefly, mice were deeply anesthetized with medetomidine 0.75 μg/kg, midazolam 0.4 μg/kg, and butorphanol 0.5 μg/kg (a mixture of three types of anesthesia), and then the tibial and common peroneal branches of the sciatic nerve were ligated with a silk suture and transected distally, whereas the sural nerve was left intact. In the sham controls, the sciatic nerve and its branches were exposed without any lesions. Following all procedures, the animals were moved to a recovery cage, observed until they were fully ambulatory and able to take food and water, and then transferred to a cage bedded with soft sawdust with free access to food and water. After the surgery, the animals were not treated with drugs to reduce hypersensitivity. The mice were handled gently to minimize and eliminate animal distress during the examination. Mice from each group were sacrificed on days 3, 7, and 21 for histological analysis (six mice for each time point).

### 2.2. Behavior Test

Mice were placed on an elevated wire grid, and following a 15 min habituation period, the lateral portion of the plantar surface of the left hind paw was stimulated using von Frey monofilaments (Touch Test; North Coast Medical Inc., Gilroy, CA, USA). A positive response was defined as a brisk withdrawal or licking of their hind paw upon stimulus presentation. The threshold was taken as the lowest force that evoked a positive response to one of five repetitive stimuli [16]. These actions were performed before SNI and before each sample was removed (on days 3, 7, and 21).

### 2.3. Tissue Preparation and DESI-MSI Data Acquisition

Mice were sacrificed with a controlled overdose of medetomidine 2.25 μg/kg, midazolam 1.2 μg/kg, and butorphanol 1.5 μg/kg (a mixture of three types of anesthesia). After that, the brain and a 1 cm segment of the spinal cord, including the L4 segment, were extracted. After dissection, the brain was immediately flash-frozen in dry ice, whereas the spinal cords were embedded in CMC solution and then flash frozen in dry ice, followed by storage at −80 °C. Tissue sections (10 μm) were cut at −20 °C using a cryostat (CM1950; Leica, Wetzler, Germany) and placed onto glass slides (Matsunami, Osaka, Japan). For consistency, the spinal sections of SNI mice and sham mice were placed onto the same slides.

Before being subjected to DESI-MSI analysis, the glass slides containing the brain and spinal cord sections were allowed to stand at 20 °C for 2–3 min to remove extra water. Thereafter, DESI-MSI data from mice samples were acquired using an OmniSpray-2D DESI ion source (Prosolia Inc., Indianapolis, IN, USA) equipped with a quadrupole time-of-flight mass spectrometer (XevoG2-XSQ-TOF; Waters, Milford, MA, USA) in the positive ion mode. Before analysis, the DESI-MSI instrument was calibrated using 0.5 mM sodium formate solution prepared in 90% 2-propanol. An ACQUITY UPLC Binary Solvent Manager (Waters, Milford, MA, USA) was used to deliver spray solvent during data acquisition. Standard 2-AG solution was dissolved in acetonitrile (3 μg/mL), and 0.3 μL of the standard solution was applied to the slides to confirm the detection of 2-AG by DESI-MSI and optimize DESI parameters for better ionization of 2-AG.

### 2.4. DESI-MSI Data Analysis

The Mass Lyn x 4. 1 and HDImaging ver 1. 4 software (Waters, Milford, MA, USA) were used to obtain DESI-MSI data. MSI raw data were converted into .imzML by the HDImaging software and then further converted to .imdx using the IMDX converter (version 1.20.0.10960; Shimadzu, Japan). The .imdx data were then used to analyze the distribution of candidate ions using the IMAGE REVEAL software (version 1.20.0.10960; Shimadzu, Kyoto, Japan). Top 1000 *m/z* peaks with a mass window of 0.02 Da were extracted from the total ion current (TIC) normalized mass spectra, and 2D ion images were constructed to visualize and analyze the spatial distribution of candidate ions. The 2-AG was identified as reported by Islam A et al. [7]. The same *m/z* peak was detected in mouse brain and standard 2-AG and confirmed. SNI and sham models were compared in this study. The IBM SPSS version 26.0(SPSS Inc., Chicago, IL, USA) software was used for statistical analysis. All values were expressed as mean ± standard deviation (SD). Differences were considered significant with *p* values less than 0.05 (two-tailed *t*-test).

## 3. Results

### 3.1. Effects of SNI on Escape Behavior in Mice

In the von Frey behavioral test, the baseline values were not different between the two groups (Figure 1). After treatment, there were also no differences between the control and SNI groups at any time, but the SNI group showed a lower threshold for escape behavior. The baseline value was obtained before treatment at 8 weeks of age; the SNI group showed a reduced threshold for behavioral triggering.

### 3.2. Distribution of 2-AG in the Hypothalamus of SNI Mice

On day 3 post-treatment, the 2-AG level in the whole brain was lower in the SNI group than in the sham group (*p* = 0.019, Figure 2A). The difference in 2-AG level was particularly pronounced in the hypothalamus, showing a decrease in the SNI group (*p* = 0.002, Figure 2A).

Similarly, on day 7 after treatment, the 2-AG level in the whole brain was lower in the SNI group than in the sham group (*p* = 0.013, Figure 2B). Furthermore, in the hypothalamus, the 2-AG level was decreased in the SNI group, similar to the result on day 3 post-treatment (*p* = 0.001, Figure 2B).

At 21 days post-treatment, the SNI group also showed a similar decrease in the 2-AG level in the whole brain (*p* = 0.032, Figure 2C). A similar trend was also observed in the hypothalamus (*p* = 0.026).

### 3.3. Distribution of 2-AG in the Periaqueductal Gray (PAG) of SNI Mice

On day 3 post-treatment, the 2-AG level in the whole brain was decreased in the SNI group (*p* = 0.019, Figure 3A). The difference was particularly pronounced in the midbrain; 2-AG levels in the superior colliculus (*p* = 0.002, Figure 3A) and reticular formation were lower in the SNI group than in the sham group (*p* = 0.004, Figure 3A). 2-AG level in the PAG was also decreased in the SNI group (*p* = 0.005, Figure 3A).

On day 7 post-treatment, there was no difference in 2-AG level in the whole brain between the SNI and sham groups (*p* = 0.070, Figure 3B). In the midbrain, however, there was a difference between the two groups; 2-AG levels in the superior colliculus (*p* = 0.036, Figure 3B) and reticular formation (*p* = 0.010, Figure 3B) were lower in the SNI group, similar to the result on day 3 post-treatment. 2-AG level in the PAG was also decreased in the SNI group (*p* = 0.024, Figure 3B).

On day 21 post-treatment, there was no difference in 2-AG level in the whole brain between the SNI and sham groups (*p* = 0.225, Figure 3C). There was also no difference in 2-AG level in the midbrain (*p* = 0.407, Figure 3C) and PAG (*p* = 0.107, Figure 3C) between the two groups.

In addition, AEA, oleoylethanolamide (OEA), and palmitoylethanolamide (PEA) were also analyzed but could not be detected in mice brains by DESI-MSI (Appendix A).

### 3.4. Distribution of 2-AG in the Lumbar Spinal Cord of SNI Mice

Compared with the sham control mice, the 2-AG level in the SNI mice’s lumbar spinal cords was increased by 1.34-fold (*p* = 0.002, Figure 4) on day 3 post-treatment and by 1.39-fold (*p* = 0.008, Figure 4) on day 7 post-treatment. On day 21, however, the increase was only 1.08-fold (*p* = 0.211, Figure 4).

## 4. Discussion

This is the first report to investigate the dynamics and distribution of 2-AG in the brain and spinal cord of a mouse model of neuropathic pain by using DESI-MSI.

Endogenous cannabinoids have been thought to exert their antinociceptive effects through CB1 receptors in the PAG, dorsal horn of the spinal cord, and dorsal root ganglion, which are the sites related to pain perception [17,18,19]. In recent years, endogenous cannabinoids have attracted attention and are considered a promising target for new therapies. However, the endogenous cannabinoids, anandamide and the 2-AG are not stored in advance, but rather produced on-demand [20]. Moreover, they are metabolized within a short period of time, making them difficult to analyze. Thus, the effects of endogenous cannabinoids on neuropathic pain are still unclear [21,22,23].

There have been numerous reports on the expression and distribution of endogenous cannabinoids and their receptors in relation to pain and irritation [24]. Walker JM et al. [25] reported that formalin administration to the hindlimb of rats induced the release of AEA to the PAG. Mitrirattanakul S et al. [26] used a neuropathic pain model of chronic constriction injury (CCI) and reported that the levels of AEA and 2-AG in the PAG were elevated on day 3 after CCI and then increased 1.3 and 3-fold, respectively, on day 7, when mechanical allodynia was maximal. It has been reported that in an L5 spinal nerve ligation model of neuropathic pain, AEA level in the spinal cord increases, but the 2-AG level does not. Others have reported increased CB1 receptor protein in the lumbar spinal cord of a spinal nerve ligation model [11] and an SNI model [15]. Therefore, endogenous cannabinoids might be closely associated with pain, especially neuropathic pain. Our study revealed that 2-AG expression decreased in the hypothalamus, midbrain, and PAG, but increased in the spinal cord during the acute and subacute phases after injury, that is, days 3 and 7 after SNI treatment. This accumulation of 2-AG in the hypothalamus, midbrain and PAG of sham mice’s brains is possibly associated with pain and perineural inflammation due to the treatment. However, the reason for decreased 2-AG levels in the hypothalamus, midbrain, and PAG of SNI mice is not clear yet. This may result in a lower pain threshold than normal, and this effect was reflected in behavioral tests. These data indicate that 2-AG, which modifies synaptic transmission efficiency and causes pain threshold changes, is decreased in the brain and increased spinal cord. The increase in 2-AG in the spinal cord, which functions to habituate to pain that is becoming chronic by lowering the pain threshold, may be meant to prevent the sending of more signals than necessary from the local area. On the other hand, a decrease in 2-AG in the brain, which is known to induce anxiety and make behavior more cautious, and to induce depression-like behavior [27], increases the probability of survival in mice by making them more cautious in their behavior after trauma, which in nature often occurs when they are attacked by foreign enemies. If we dare to extrapolate these findings to humans, we can consider that they can be interpreted as the molecular background of the situation of lack of energy after injury. In this study, we also tried to detect AEA, and other endocannabinoids-like substances, including OEA and PEA, were not detected by DESI-MSI.

Islam A et al. found that water immersion stress increased 2-AG levels in the hypothalamus, hindbrain, and midbrain, which returned to basal levels when the stress load was removed [7]. In this study, the stress of neuropathic pain that we inflicted on the mice showed a decrease in the 2-AG distribution in that region. Whether this is a response specific to neuropathic pain or whether 2-AG is depleted due to too much stress is unknown and requires further investigation. In particular, since we used the SNI model in this study and stress could not be removed. So, it is unclear what the dynamics of 2-AG will be in the absence of stress, which also requires further investigation.

The pathway of somatosensory transmission to the brain is divided into (1) the lateral system, in which nerve fibers from neurons in the dorsal horn of the spinal cord cross the midline of the spinal cord and ascend the opposite side, passing through the brainstem and thalamus to the somatosensory cortex, where sensory reception occurs; and (2) the medial system, which is involved in emotional cognition associated with pain. The thalamus is the entry point for the reception of somatosensory, visual, auditory, and gustatory information, which is processed in the somatosensory cortex, one of the sensory areas in the cerebral cortex. Pain information is also sent to the hypothalamus, causing autonomic nervous system reactions such as sweating and palpitations. Prolonged pain further stimulates the limbic system, including the amygdala, which is involved in the expression of fear and emotion; the hippocampus, which is involved in memory formation; and the prefrontal cortex, which controls decision-making and motor behavior; as well as the associated areas, which are involved in complex integrated brain functions. The PAG plays a particularly important role in pain control because it sends nociception information to the limbic system and prefrontal cortex. Based on the processed information, it overrides the previous pain experience and reflects it in memory [28].

In addition, to suppress excessive pain input, the pain control mechanism includes a descending pain suppression system. This is a pathway where neurons are excited, descending from the PAG and projected to the dorsal horn of the spinal cord to suppress pain. Our study suggests that this pathway can also be disrupted, indicating that both ascending and descending pain control functions are disrupted in the SNI mouse model.

Our findings showed that 2-AG expression in the hypothalamus was decreased at the chronic stage after injury (21 after SNI treatment). However, there was no difference in 2-AG expression in the PAG, its surroundings, or the lumbar spinal cord. This is consistent with the report, where the authors reported that there was no difference in 2-AG level in the PAG or spinal cord of male mice at day 42 after SNI treatment [29]. However, the authors did not examine the 2-AG level in the hypothalamus.

There are some limitations in our present study. One is that this study did not examine the 2-AG level in the dorsal root ganglion, which is thought to be involved in peripheral pain control. It has been suggested that 2-AG is present in the lumbar medullary ganglion and plays a role in pain control. Future studies should examine the response of other regions. Another limitation is that the distribution of 2-AG was investigated only in the acute, subacute, and chronic phases after injury and not in the hyperacute phase, such as immediately after injury. The decrease in 2-AG level in the acute phase may be due to a mechanism in the hyperacute phase, and thus further investigation is required. Additionally, a negative control could help to understand the relation between 2-AG and neuropathic pain. However, in this study, we did not have a negative control.

## 5. Conclusions

This study revealed that compared to sham-operated mice, SNI mice exhibited decreased 2-AG expression during the acute and subacute phases after injury, particularly in the hypothalamus and PAG. This finding suggests that 2-AG is associated with a lower pain threshold under neuropathic pain conditions. Conversely, 2-AG expression was increased in the spinal cord medulla, suggesting the importance of 2-AG in pain control. Therefore, 2-AG may be a potential target for neuropathic pain treatment.

## Figures and Tables

**Figure 1 cells-11-04130-f001:**
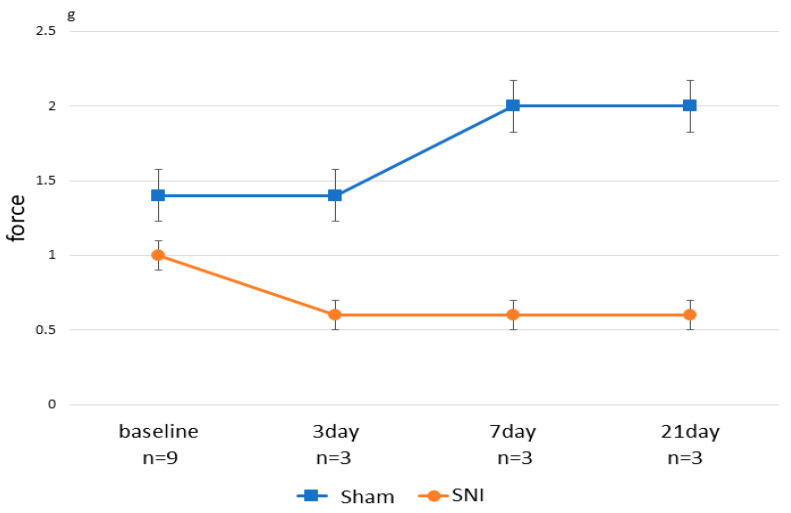
Time series results of the von Frey behavior test. The vertical axis represents the minimum force (g) at which mice responded positively in behavioral tests. The baseline is the value of behavioral tests before SNI or Sham treatment. Behavioral tests on Days 3, 7, and 21 were performed before the sacrifice, respectively.

**Figure 2 cells-11-04130-f002:**
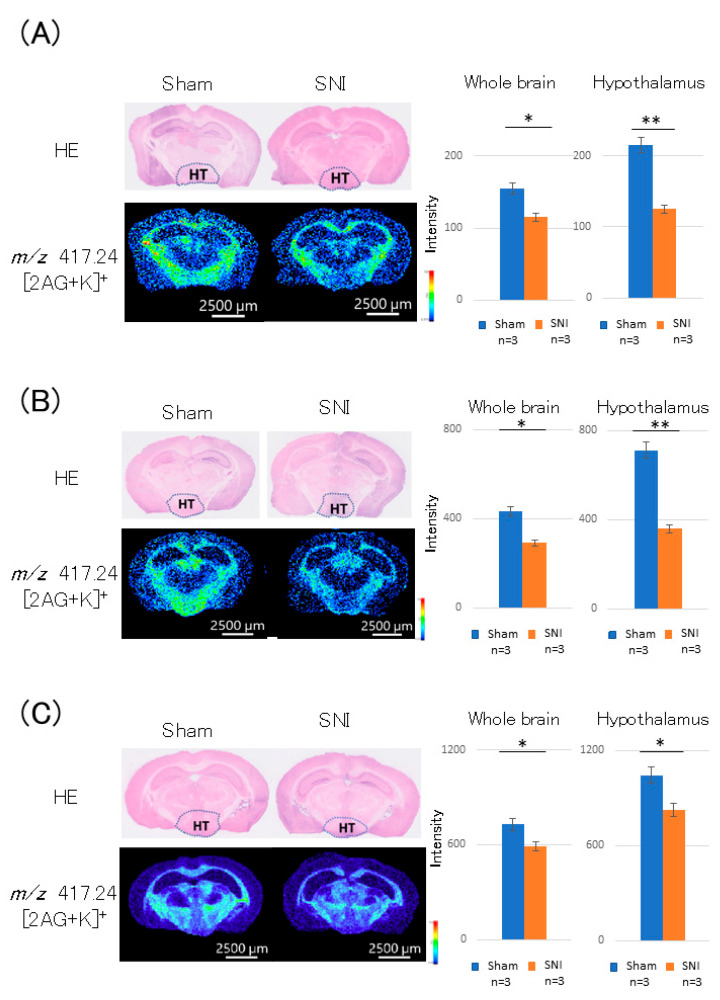
Distribution of 2-AG at the thalamic level in the brain cross-sections of SNI mice. (**A**) On day 3 post-treatment, 2-AG level in the whole brain was reduced in the SNI group. The difference in 2-AG level was particularly pronounced in the hypothalamus, showing a decrease in the SNI group. (**B**) On day 7 post-treatment, 2-AG level in the whole brain was reduced in the SNI group. Furthermore, 2-AG level in the hypothalamus was decreased in the SNI group, similar to the result on day 3 post-treatment. (**C**) On day 21 post-treatment, the SNI group also showed a similar decrease in the 2-AG level, but the difference was smaller. A similar trend was observed in the hypothalamus. *: *p* < 0.05, **: *p* < 0.01.

**Figure 3 cells-11-04130-f003:**
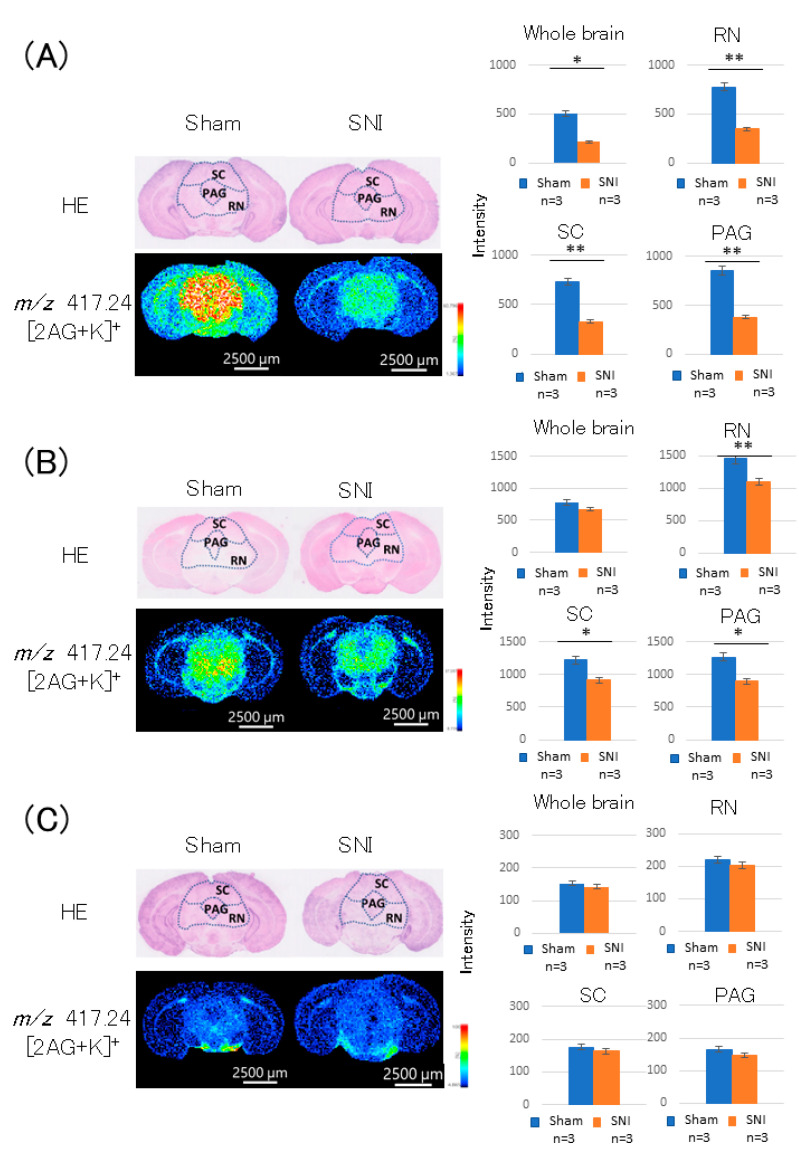
Detection of 2-AG in the PAG region of mice brains. (**A**) On day 3 post-treatment, 2-AG level in the whole brain was reduced in the SNI group. The difference was particularly pronounced in the midbrain; 2-AG levels in the superior colliculus (SC) and reticular formation (RN) were lower in the SNI group than in the sham group. 2-AG level in the PAG was also decreased in the SNI group. (**B**) On day 7 post-treatment, 2-AG level in the whole brain was reduced in the SNI group. In the midbrain, however, there was a difference in 2-AG between the two groups: 2-AG levels in the SC and RN were lower in the SNI group than in the sham group, similar to the result on day 3 post-treatment. 2-AG level in the PAG was also decreased in the SNI group. (**C**) On day 21 post-treatment, there was no difference in 2-AG level in the whole brain between the SNI and sham groups. There was also no difference in 2-AG levels in the midbrain and PAG between the two groups. *: *p* < 0.05,**: *p* < 0.01.

**Figure 4 cells-11-04130-f004:**
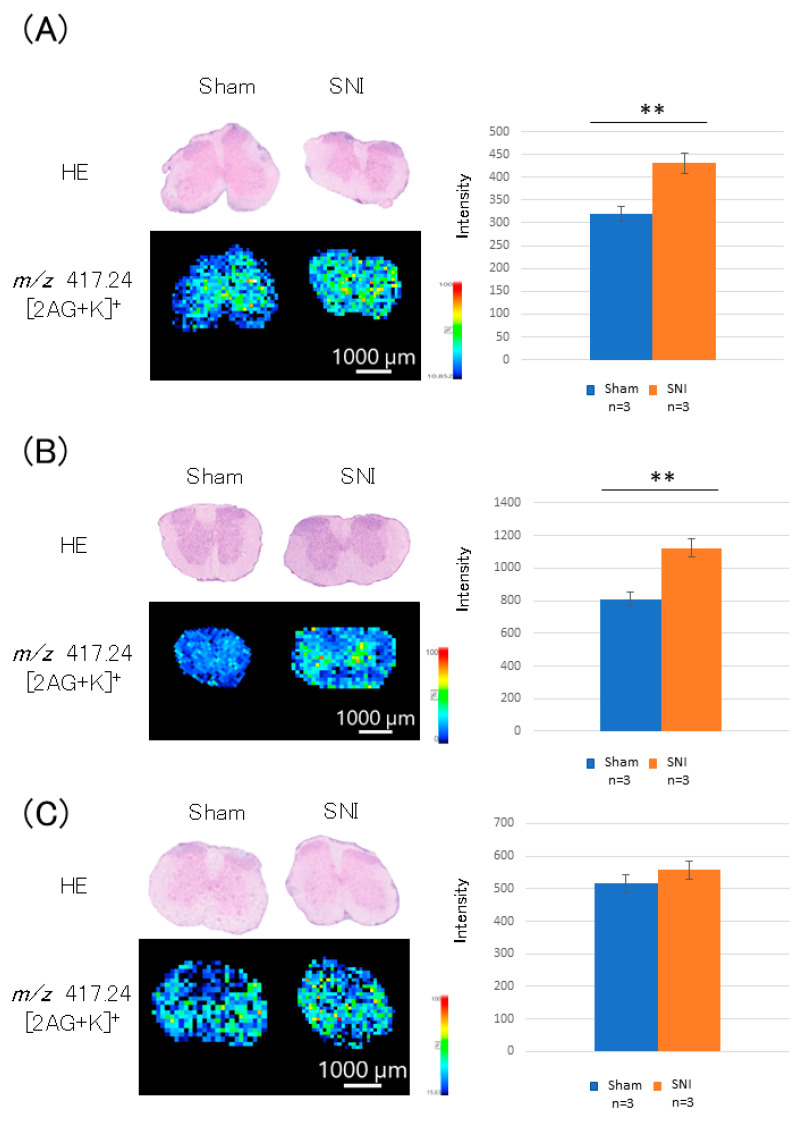
2-AG distribution in the lumbar spinal cord cross-section. (**A**) On day 3 post-treatment, 2-AG level was increased by 1.34-fold (*p* = 0.002) in the SNI group compared to the sham group. (**B**) On day 7 post-treatment, 2-AG level was increased by 1.39-fold (*p* = 0.008) in the SNI group compared to that in the sham group. (**C**) On day 21 post-treatment, 2-AG level increased by 1.08-fold in the SNI group compared to that in the sham group (*p* = 0.211). *: *p* < 0.05,**: *p* < 0.01.

## Data Availability

Not applicable.

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
