# Peer review of "Expression and Kinetics of Endogenous Cannabinoids in the Brain and Spinal Cord of a Spare Nerve Injury (SNI) Model of Neuropathic Pain"

_cells, 2022, doi:10.3390/cells11244130_

Round 1

Reviewer 1 Report

This is a very good research article, it is very important. This article is about 2-AG, which has antinociceptive properties. It can be detected by DESI-MSI. In a SNI-model, 2-AG was detected in different parts of the brain and the spinal cord. The presentation of the results in tables, figures is very good. The results and the discussion are very comprehensibles and well written. The English language is very correct. The list of references is very well chosen. However, it would be good to add a brief paragraph about the mechanism of action of 2-AG.

I recommend a minor revision.

Author Response

Dear Reviewer,

Thank you for your suggestion, we have made the following changes and additions to the text regarding the details of 2-AG.

Line 41;  Initially, anandamide was studied first, but 2-AG was found to be 50-200 times more abundantly expressed in the brain than anandamide [5], and 2-AG is considered a true bioactive substance because it is a full agonist of the CB1 receptor. 2-AG is also synthesized from phospholipid precursors in postsynaptic neurons by activation of phospholipase C (PLC) and diacylglycerol lipase (DAGL). Subsequently, when 2-AG is released at the synaptic cleft, it is returned into the cell and degraded by hydrolytic enzymes such as fatty acid amide hydrolase (FAAH) and monoacylglycerol lipase (MAGL) present in the presynaptic cell.

Thank you for all reviewer and editor. We hope these changes meet your requirements.

Reviewer 2 Report

This is a well-conducted and well-illustrated study. The statistical analyses are appropriate.

I recommend some minor corrections, as follows.

Line 20 - please correct.

37 - initial capitalisation is not required for anandamide.

42 - the sentence "It is ... progressing" does not make sense.

Author Response

Dear Reviewer,

Thank you for your suggestion. We have proofread the English text and made significant changes to the English wording.

We have accordingly made the following changes to the points you pointed out.

〇Line 20 - please correct.:

The expression and distribution of 2-AG in the brain and spinal cord of a spare nerve injury (SNI) model of neuropathic pain was studied using DESI-MSI.

Line 20;  In this study, the expression and distribution of 2-AG in the brain and spinal cord of a spare nerve injury (SNI) model of neuropathic pain was examined using DESI-MSI.

〇37 - initial capitalisation is not required for anandamide.

Anandamide

Line 38;  anandamide

〇42 - the sentence "It is ... progressing" does not make sense.

I believe you are correct.

I have removed it from the text.

Thank you for all reviewer and editor. We hope these changes meet your requirements.

Reviewer 3 Report

The manuscript "Expression and kinetics of endogenous cannabinoids in brain and spinal cord of a spare nerve injury (SNI) model of neuropathic pain." describes the determination of 2-AG by DESI-MSI in brain and spinal cord of mice comparing a SNI and a sham group. The authors state that this is the first application of DESI-MS in this context which is right. However, the information provided in the manuscript are very basic and not much more than a description of the findings without detailed interpretation. The whole discussion is more about the background and just in a very short section about the data - this has to be improved. Furthermore, the manuscript suffers from poorly written English making it very difficult to follow. All in all I cannot recommend this manuscript for publication. However, the study itself is of interest but needs a better written manuscript and a more detailed interpretation of the data. Furthermore, it might be helpful to have a look at other endocannabinoids and endocannabinoid like substance as for example PEA and OEA (which should be detectable as the concentrations are comparable to the concentration of 2-AG).

Major and minor points:

·         2-AG ist not “believed to have bioactivity” – this was proven in hundreds of publications!

·         Introduction is missing a red thread - please think about a better and more informative structure

·         Which (local) authorities did approve this animal test

·         Its not becoming clear from the methods section which groups have been compared

·         How was the quantification of 2-AG performed? Whats the resolution the authors used?

·         How do the authors explain the non significant difference between the two groups when comparing escape behavior

·         Identification of 2-AG from the ms-data must be clarified – please add a detailed description to the methods part (mass is just listed in the figures but not in the manuscript itself)

Author Response

Dear Reviewer,

Thanks for your interesting input.

I have submitted it to Editage, a company specializing in English editing, and received substantial revisions.

I have also added the following acknowledgments for that.

Line 311;  I would like to thank Editage (www.editage.com) for English language editing.

As for your suggestion about the measurement of PEA and OEA, I will take it up as an issue for future research.

  • 2-AG ist not “believed to have bioactivity” – this was proven in hundreds of publications!

As you said, the bioactivity of 2-AG has already been elucidated after various studies. Therefore, I have corrected this sentence as follows

The relationship between endogenous cannabinoids and neuropathic pain has been actively studied, and 2-arachidonoyl glycerol (2-AG), which is believed to have bioactivity, is one of the lipids that has received the most attention.

Line 15;  The role of endogenous cannabinoids in neuropathic pain has been actively studied, among which 2-arachidonoyl glycerol (2-AG) has received the most attention.

  • Introduction is missing a red thread - please think about a better and more informative structure

I think you are right that I am missing some information on INTRODUCTION.

Therefore, I have mentioned the metabolism of 2-AG.

Initially, research on anandamide preceded, but 2-AG was 50-200 times more ex-pressed in the brain than anandamide [5], and because it is a complete agonist of the CB1 receptor, it is a more true bioactive substance.

Line 41;  Initially, anandamide was studied first, but 2-AG was found to be 50-200 times more abundantly expressed in the brain than anandamide [5], and 2-AG is considered a true bioactive substance because it is a full agonist of the CB1 receptor. 2-AG is also synthesized from phospholipid precursors in postsynaptic neurons by activation of phospholipase C (PLC) and diacylglycerol lipase (DAGL). Subsequently, when 2-AG is released at the synaptic cleft, it is returned into the cell and degraded by hydrolytic enzymes such as fatty acid amide hydrolase (FAAH) and monoacylglycerol lipase (MAGL) present in the presynaptic cell.

  • Which (local) authorities did approve this animal test

This is stated in the Institutional Review Board Statement, but as you said, it is difficult to understand, so I have also added the following sentence in the 2. Materials and Methods text.

Line 75;  This experimental plan was reviewed and approved by the Animal Experiment Committee of Hamamatsu University School of Medicine (2020085).

  • Its not becoming clear from the methods section which groups have been compared

As you said, I did not explain enough about the groups we are comparing.

Therefore, I have clearly stated in 2. Materials and Methods text. that I am comparing SNI and sham models.

Line 142;  SNI and sham models were compared.

  • How was the quantification of 2-AG performed? Whats the resolution the authors used?

Sorry, we did not perform quantitative analysis. We have done the qualitative analysis, and compared between/among groups based on the average signal intensity of 2AG.

And in our MSI data, we used 100 µm x 100 µm pixel size (spatial resolution).

  • How do the authors explain the non significant difference between the two groups when comparing escape behavior

As you said, the results of this behavior test did not reach statistical significance. However, there was a difference in the results of the behavior test between the two groups. I believe that this is because only 3 mice were compared in each schedule, and therefore the difference did not reach statistical significance.

  • Identification of 2-AG from the ms-data must be clarified – please add a detailed description to the methods part (mass is just listed in the figures but not in the manuscript itself)

Thank you for pointing this out.

We have identified 2-AG based on our recent study (Ariful Islam et al., 2022, BBRC). Therefore, we have noted that as follows.

Line 140;  2-AG was identified as reported by Ariful et al [7]. The same m/z peak was detected in mouse brain and standard 2-AG and confirmed by MS/MS.

Thank you for all reviewer and editor. We hope these changes meet your requirements.

Reviewer 4 Report

The authors investigated the distribution of 2-AG in a mouse model of neuropathic pain using DESI imaging mass spectrometry. They found that 2-AG expression was decreased in the brain areas that regulate the sensation of pain and suggested that an insufficient amount of 2-AG may lower a pain threshold and contribute to causing neuropathic pain.

The finding is novel and interesting. The present study offers helpful insight into the neurological mechanisms for medically-intractable pain. Despite the imaging endocannabinoids being challenging, the authors tactfully accomplished the difficult task by using DESI in clinical application. The figures are comprehensive and well-organized. Results are described clearly point by point. There is no problem found in the scientific content of the manuscript.

However, the manuscript was written in a complicated style and was hard to read. In particular, the introduction, Discussion, and Conclusion do not communicate well. It is unfortunate that the scientifically-excellent paper loses readers’ attention and patience so quickly. The paper can be improved a lot if language assistance is available. The authors are advised to work with native English-speaking scholars in revising the manuscript or consider using language assistance offered by the MDPI manuscript submission system.

Author Response

Dear Reviewer,

Thank you for pointing this out.

As you stated, the English was not complete.

I have submitted it to Editage, a company specializing in English editing, and received substantial revisions.

I have also added the following acknowledgments for that.

Line 311;  I would like to thank Editage (www.editage.com) for English language editing.

Thank you for all reviewer and editor. We hope these changes meet your requirements.

Round 2

Reviewer 4 Report

The manuscript is well revised. I feel that not all corrections are highlighted, but not a major issue.

Author Response

Dear Reviewer,

Thanks for your comments.

I had made a mistake in the way I did the highlighting.

I have corrected it in the correct way this time.

Thank you for all reviewer and editor. We hope these changes meet your requirements.

Additionally,WS files are also attached.
